

# A public dataset of running biomechanics and the effects of running speed on lower extremity kinematics and kinetics

Reginaldo K. Fukuchi[1], Claudiane A. Fukuchi[2] and Marcos Duarte[1,2]

[1] Biomedical Engineering Program, Universidade Federal do ABC, São Bernardo do Campo, São Paulo, Brazil
[2] Neuroscience and Cognition Graduate Program, Universidade Federal do ABC, São Bernardo do Campo, São Paulo, Brazil

## ABSTRACT

**Background**. The goals of this study were (1) to present the set of data evaluating running biomechanics (kinematics and kinetics), including data on running habits, demographics, and levels of muscle strength and flexibility made available at Figshare (DOI: 10.6084/m9.figshare.4543435); and (2) to examine the effect of running speed on selected gait-biomechanics variables related to both running injuries and running economy.

**Methods**. The lower-extremity kinematics and kinetics data of 28 regular runners were collected using a three-dimensional (3D) motion-capture system and an instrumented treadmill while the subjects ran at 2.5 m/s, 3.5 m/s, and 4.5 m/s wearing standard neutral shoes.

**Results**. A dataset comprising raw and processed kinematics and kinetics signals pertaining to this experiment is available in various file formats. In addition, a file of metadata, including demographics, running characteristics, foot-strike patterns, and muscle strength and flexibility measurements is provided. Overall, there was an effect of running speed on most of the gait-biomechanics variables selected for this study. However, the foot-strike patterns were not affected by running speed.

**Discussion**. Several applications of this dataset can be anticipated, including testing new methods of data reduction and variable selection; for educational purposes; and answering specific research questions. This last application was exemplified in the study's second objective.

Corresponding author
Reginaldo K. Fukuchi,
regifukuchi@gmail.com,
reginaldo.fukuchi@ufabc.edu.br

## INTRODUCTION

Long-distance running has become a very popular form of physical activity among individuals pursuing a healthy lifestyle (*Stamatakis & Chaudhury, 2008*). The health benefits of regular running are well known, however worrisome rates of running-related injuries have been reported and have associated burdens and economic costs (*Hespanhol Jr et al., 2016*).

Running biomechanics has been claimed to be associated with both running injury etiology (*Hreljac, 2004*) and running economy (*Moore, 2016*). Impact forces, foot pronation and shoes have all been linked to injuries although the literature is inconclusive about their role in the risk of running injuries (*Nigg et al., 2015*). Running foot strike patterns have also

been the focus of great interest in the discussion pertaining to biomechanical injury factors which has resulted in an increased number of studies examining their effects on the rate of injuries and on running biomechanics (*Daoud et al., 2012*; *Hall et al., 2013*). Another factor that has been related to running injuries is the excessive pace or excessive training volume (*Nielsen et al., 2013*). However, only a handful of studies have focused on examining the effect of running speed on gait biomechanics (*Petersen et al., 2014*; *Schache et al., 2011*), and the available evidence is rather conflicting. This can be partly explained by the fact that running biomechanics has been examined either without controlling the gait speed or by obtaining the data for a single controlled gait speed. In addition, although these studies added new data, they typically used small sample sizes and limited sets of biomechanical variables and considered only one part of the gait cycle (either the stance or swing phase), not to mention that the raw data from these studies are typically not freely available. Therefore, there is a need for studies that examine a larger set of runners across a range of gait speeds and that consider a larger set of biomechanical variables (e.g., kinematics and kinetics).

Although a study including these features would greatly contribute to advancing knowledge about the effect of gait speed, some challenges are likely to be encountered. The complex, multivariate nature of biomechanics data challenges traditional data-analysis methods and, therefore, limits the ability of clinical-gait researchers to interpret these results and apply this knowledge to intervention procedures. To overcome these challenges and encourage the development of innovative tools that can address the nature of gait-biomechanics data, data sharing has been advocated (*Ferber et al., 2016*). Unfortunately, there are few publicly available datasets in the human movement science area (see, for example, *Moore, Hnat & Van den Bogert, 2015*; *Santos & Duarte, 2016*). In fact, to our knowledge, there is no running biomechanics data sets with varying gait speeds available to the public. Therefore, a public data set of raw running biomechanics data would address this limitation and would welcome international research groups to use this data set to provide further insights about the related changes in biomechanics under varying running speed conditions. Therefore, the purposes of this study were (1) to present the set of raw and processed data on running biomechanics made available at Figshare (DOI: 10.6084/m9.figshare.4543435); and (2) to examine the effect of running speed on selected gait-biomechanics variables associated with both injury etiology and running performance.

## MATERIALS AND METHODS

This study aimed to examine the effect of running speed on selected gait-biomechanics variables and to make the resulting dataset available in a public repository. The study was conducted at the Laboratory of Biomechanics and Motor Control (BMClab; http://demotu. org) at the Federal University of ABC (UFABC). The data collection was performed by experienced physiotherapist researchers. A pilot study with five subjects was conducted prior to the beginning the principal study. This study was approved by the local ethics committee of the UFABC (CAAE: 53063315.7.0000.5594), and written, informed consent was obtained from each subject prior to participation in the study. The data collection was designed to record the following measurements, which are described in detail later:

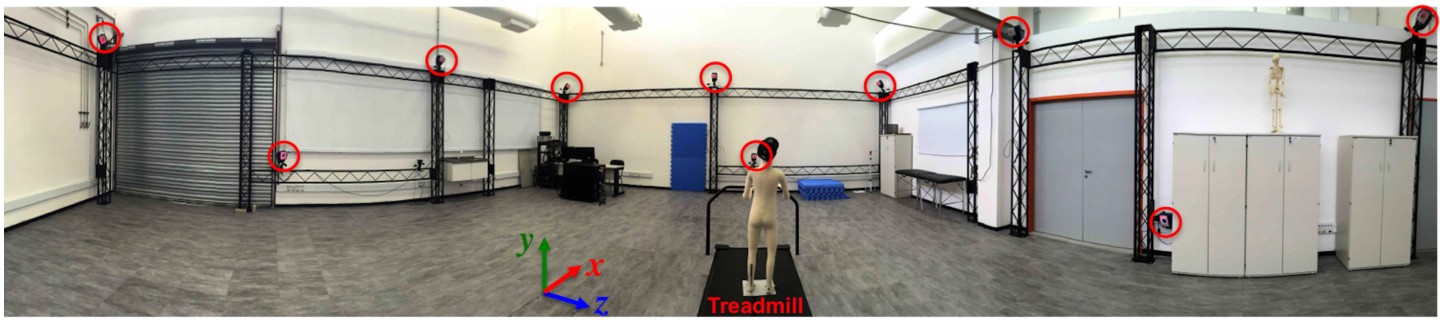

**Figure 1** **Overview of the Laboratory of Biomechanics and Motor Control.** Expanded view of the Laboratory of Biomechanics and Motor Control (BMClab), showing 10 of the 12 motion-capture system cameras (marked with red circles), the instrumented treadmill, and the laboratory-coordinate system.

three-dimensional (3D) kinematics of the two lower limbs and pelvis, ground-reaction forces (GRF) during running on a treadmill at various speeds, and the strength and flexibility of selected muscle groups and joints.

## Participants

The study analyzed a convenience sample of 28 subjects who were recruited through posted flyers, advertisement on the BMClab Internet home-page (http://demotu.org), and social media. The inclusion criteria included being a regular runner with a weekly mileage greater than 20 km, a minimum average running pace of 1 km in 5 min during 10-km races, and familiarity and comfort with running on a treadmill. The exclusion criteria were the presence of any neurological or musculoskeletal disorder that compromises its locomotion or the use of any assistive devices. The data related to demographics, running-training characteristics, previous injuries, and other relevant information were made available in the public dataset (see also the Table S1).

## Equipment

The running kinematics were collected via a 3D motion-capture system with 12 cameras having 4 Mb of resolution and the Cortex 6.0 software (Raptor-4, Motion Analysis, Santa Rosa, CA, USA). The GRF data were collected via an instrumented, dual-belt treadmill (FIT, Bertec, Columbus, OH, USA).

The cameras were distributed around the laboratory such that they aimed at the instrumented treadmill's motion-capture volume (Fig. 1). The cameras were mounted in a metallic truss setup structure with a length of 11.5 m, a width of 9.3 m, and a height of 2.8 m. This structure allowed positioning some cameras with varying elevations; however, most were placed atop the truss setup to optimize capturing the markers during the running trials (Fig. 1). The instrumented treadmill was mounted over a pit, with the treadmill surface at the same level as the laboratory floor (Fig. 1).

The Cortex 6.0 software (Motion Analysis, Santa Rosa, CA, USA) was used to (1) calibrate the motion-capture volume; (2) capture and identify the reflective markers; and (3) prepare the data and export it to the c3d file format. To provide an unbiased, raw

dataset having marker trajectories and force signals, no further processing (e.g., filtering, gap filling) was performed on the data.

The motion-capture volume consisted of an area 3.1 m long, 2.3 m wide, and 1.2 m high, and this volume was calibrated daily. The system was deemed properly calibrated only if the length of the calibration wand, which was measured by the capture system, was within 0.10 mm of the true wand length. The rates of acquisition of the kinematics and kinetics data were set at 150 Hz and 300 Hz, respectively.

The laboratory-coordinate system used for the study was the same as that proposed by the International Society of Biomechanics (*Wu & Cavanagh, 1995*) and, as shown in Fig. 1, contained the following:

- $X$-axis in the direction of gait progression and positive pointing forward.
- $Y$-axis in the vertical direction and positive pointing upward.
- $Z$-axis in the medial-lateral direction and positive pointing to the right.

To record the strength and flexibility measures of selected muscle groups and joints, a hand-held dynamometer (HHD) (range: 0–1,330 N; accuracy: ±1%; resolution: 1 N; Nicholas MMT, Lafayette Instruments, Lafayette, IN, USA) and a magnetic-angle locator (Model 700; Johnson Level & Tool Mfg. Co., Inc., Mequon, WI, USA) were used, respectively.

## Protocol

The data-collection protocol involved the following procedures:

1. Interview. Upon arrival, the participant was introduced to the laboratory and given a brief explanation of the experimental procedures. Then, the participant was asked to provide written informed consent and undergo a brief interview regarding eligibility criteria, demographic data, and running habits.
2. Preliminary measurements. Body height and mass were measured, and shoe-fitting was conducted to determine the appropriate shoe size. All participants wore neutral laboratory-controlled shoes (Nike Dual Fusion X).
3. Marker placement. The study used 48 technical and anatomical reflective markers (see details in Table 1 and Fig. 2). Clusters with four technical markers, placed in a rigid shell, were used on the thigh and shank segments. Their design was based on *Cappozzo et al. (1997)*. These shells were securely fastened to the segments using a combination of elastic and Velcro straps.
4. Standing calibration trial. A template was used to align the subject's feet in a standardized position such that the long axes of the feet were parallel to the $X$-axis of the laboratory-coordinate system (Fig. 2). Then, the markers' 3D coordinates were recorded for 1 s.
5. Removal. After the calibration trial, the anatomical markers were removed except for those considered both anatomical and technical markers (T/A in Table 1).
6. The force plates were zeroed, the subject was asked to step onto the treadmill, and the following protocol was followed:
   a. The subject walked at 1.2 m/s for 1 min to become familiar with the treadmill.

b.  Next, the subject was asked to stay on the left belt of the treadmill, the belt speed was incrementally increased to 2.5 m/s, and after a 3-min accommodation period at this velocity, the data were recorded for 30 s. This procedure was repeated at speeds of 3.5 m/s and 4.5 m/s, always in the same sequence.

c.  After the running trials, the treadmill speed was again set to 1.2 m/s for a 1-min cool-down period prior to being stopped.

7.  Measuring the flexibility of the iliotibial band using the angle locator during the Ober's test procedure. In brief, the test is performed with the subject lying on his/her side. The examiner then passively moves the tested leg (leg on top) into hip flexion, abduction, and extension and lowers the limb into adduction until it stops, limited by soft tissue stiffness. Further details about this test procedure can be found in *Fukuchi et al. (2014)*.

8.  Measuring the flexibility of the hip flexors using the angle locator during the Thomas' test procedure. In brief, the test is performed while the subject lies supine with the hip joint positioned over the edge of the examination table and flexes the contralateral limb (hip and knee), bringing the thigh to the chest and holding it while the contralateral leg is suspended by the resistance imposed by the soft tissue to withhold the limb's weight. Further details about this test procedure can be found in *Fukuchi et al. (2014)*.

9.  Three trials of maximal isometric voluntary contraction (MIVC) of the hip abductors, extensors, and internal and external rotator muscles were measured. The procedures used to take these measurements were described previously (*Fukuchi et al., 2014*).

The definition of the anatomical-segment coordinate system used to determine the 3D position and orientation of the lower extremity and pelvis segments was a combination of anatomical-frame conventions proposed previously (*Cappozzo et al., 1995*; *Fukuchi et al., 2014*). A model template file (RBDSmodelV3D.mdh) for the Visual 3D software (C-Motion Inc., Germantown, MD, USA) is available at Figshare. This .mdh file is an ASCII file containing the definitions of all landmarks, segments, and segment properties adopted by the present study.

## Data processing and analysis

Raw marker-trajectory data and GRF data were filtered using a fourth-order, low-pass Butterworth filter with the same cut-off frequency of 10 Hz (*Kristianslund, Krosshaug & Van den Bogert, 2012*). The foot strike and toe off were determined when the vertical GRF crossed a 20-N threshold level. The foot strike patterns were determined using the strike index, which was calculated as the ratio of the center of pressure (COP) position in relation to the heel position, at foot strike, and the length of the foot. The measurements were taken, however, during instrumented treadmill running instead of on an overground condition as originally proposed by *Cavanagh & Lafortune (1980)*.

In addition, 3D hip, knee, and ankle angles were calculated using Cardan angles, with the distal segment expressed relative to the proximal segment and adopting the following convention: the first rotation described occurred in the medial-lateral axis ($Z$-axis, perpendicular to the sagittal plane), which defines the flexion-extension movement; the third rotation described was around the longitudinal axis ($Y$-axis, perpendicular to the transverse plane), which defines the internal/external rotations; and the second rotation described

**Table 1  Details of the 48 anatomical (A) and technical (T) reflective markers used to determine the position and orientation of the body segments during treadmill running.** The marker labels are consistent with those stored in files in the c3d format and with the headers of the ASCII marker files.

| # | Label | Type | Name |
|---|-------|------|------|
| 1 | R.ASIS | T/A | Right anterior superior Iliac spine |
| 2 | L.ASIS | T/A | Left anterior superior Iliac spine |
| 3 | R.PSIS | T/A | Right posterior Iliac spine |
| 4 | L.PSIS | T/A | Left posterior Iliac spine |
| 5 | R.Iliac.Crest | T | Right Iliac crest |
| 6 | L.Iliac.Crest | T | Left Iliac crest |
| 7 | R.Thigh.Top.Lateral | T | Right thigh top lateral marker |
| 8 | R.Thigh.Bottom.Lateral | T | Right thigh bottom lateral marker |
| 9 | R.Thigh.Top.Medial | T | Right thigh top medial marker |
| 10 | R.Thigh.Bottom.Medial | T | Right thigh bottom medial marker |
| 11 | R.Shank.Top.Lateral | T | Right shank top lateral marker |
| 12 | R.Shank.Bottom.Lateral | T | Right shank bottom lateral marker |
| 13 | R.Shank.Top.Medial | T | Right shank top medial marker |
| 14 | R.Shank.Bottom.Medial | T | Right shank bottom medial marker |
| 15 | R.Heel.Top | T/A | Right heel top |
| 16 | R.Heel.Bottom | T/A | Right heel bottom |
| 17 | R.Heel.Lateral | T | Right heel lateral |
| 18 | L.Thigh.Top.Lateral | T | Left thigh top lateral marker |
| 19 | L.Thigh.Bottom.Lateral | T | Left thigh bottom lateral marker |
| 20 | L.Thigh.Top.Medial | T | Left thigh top medial marker |
| 21 | L.Thigh.Bottom.Medial | T | Left thigh bottom medial marker |
| 22 | L.Shank.Top.Lateral | T | Left shank top lateral marker |
| 23 | L.Shank.Bottom.Lateral | T | Left shank bottom lateral marker |
| 24 | L.Shank.Top.Medial | T | Left shank top medial marker |
| 25 | L.Shank.Bottom.Medial | T | Left shank bottom medial marker |
| 26 | L.Heel.Top | T/A | Left heel top |
| 27 | L.Heel.Bottom | T/A | Left heel bottom |
| 28 | L.Heel.Lateral | T | Left heel lateral |
| 29 | R.GTR | A | Right greater trochanter |
| 30 | R.Knee | A | Right knee |
| 31 | R.Knee.Medial | A | Right knee medial |
| 32 | R.HF | A | Right head of fibula |
| 33 | R.TT | A | Right tibial tuberosity |
| 34 | R.Ankle | A | Right ankle |
| 35 | R.Ankle.Medial | A | Right ankle medial |
| 36 | R.MT1 | A | Right 1st metatarsal |
| 37 | R.MT5 | A | Right 5th metatarsal |
| 38 | R.MT2 | A | Right 2nd metatarsal |
| 39 | L.GTR | A | Left greater trochanter |
| 40 | L.Knee | A | Left knee |

**Table 1** (*continued*)

| # | Label | Type | Name |
|---|---|---|---|
| 41 | L.Knee.Medial | A | Left knee medial |
| 42 | L.HF | A | Left head of fibula |
| 43 | L.TT | A | Left tibial tuberosity |
| 44 | L.Ankle | A | Left ankle |
| 45 | L.Ankle.Medial | A | Left ankle medial |
| 46 | L.MT1 | A | Left 1st metatarsal |
| 47 | L.MT5 | A | Left 5th metatarsal |
| 48 | L.MT2 | A | Left 2nd metatarsal |

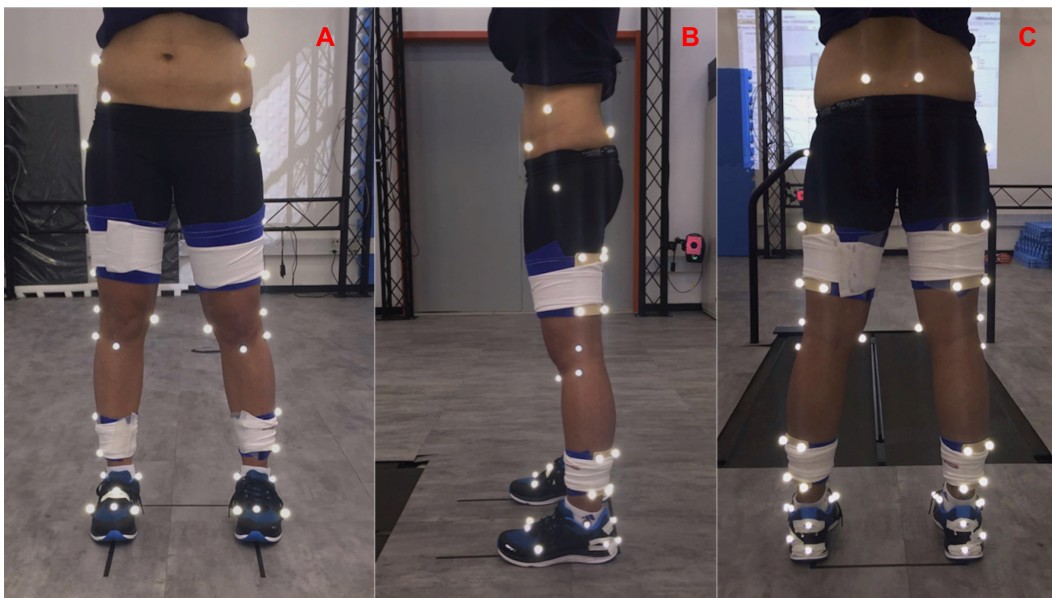

**Figure 2** **Marker set protocol.** The technical and anatomical marker set protocol during an anatomical calibration trial in the anterior (A), lateral (B) and posterior (C) views.

was around an axis perpendicular to the previous two axes, which in the anatomic position represents the anterior-posterior axis ($X$-axis, perpendicular to the frontal plane), where abduction/adduction occurs. This convention is defined simply as the $Z - X - Y$ convention and is frequently used to describe lower extremity rotations (*Cappozzo et al., 1995*). The net internal joint torques were represented in the joint-coordinate system (*Schache & Baker, 2007*) and were calculated using a standard inverse-dynamics approach. Joint powers were calculated as the scalar product of joint torques and joint angular velocities. The joint kinetic and the GRF variables were normalized by the subject's body mass.

Joint angles, joint torques, joint powers, and GRF were normalized to the gait cycle over 101 time points. Then, these curves were averaged across trials, resulting in one curve comprising each subject's average pattern. The number of footfalls varied with subject and speed, but the minimum number was always greater than 30. Next, to enable statistical comparison, discrete variables were calculated for each curve. Global maximum and minimum values for the joint angles and joint torques curves, GRF impulses, and joint

work were considered for further analysis. The GRF impulses and joint work were calculated as the area under the GRF-time and joint power-time curves, respectively. The stride length and cadence were also calculated as the spatiotemporal gait parameters. These variables have been examined in previous studies related to running biomechanics in the context of injury etiology, running performance and aging (*Fukuchi & Duarte, 2008*; *Fukuchi et al., 2011*; *Fukuchi et al., 2014*; *Fukuchi et al., 2016*; *Hall et al., 2013*). The Visual 3D software program (C-motion Inc., Germantown, MD, USA) was used to filter the marker and GRF data and to calculate joint kinematics and kinetics. Finally, these time-normalized data were exported as ASCII text files. Then, the discrete variables, GRF impulses, and joint work were calculated using in-house algorithms written in Matlab 9.0 2016a (Mathworks Inc., Natick, MA, USA). A file written for the Visual 3D software program (RBDSpipelineV3D.v3s) is available at Figshare. This file is in text format and contains a series of pipeline Visual 3D commands that were used to process the raw c3d files, which are also available at Figshare. In addition to the raw c3d files, the time-normalized kinematic and kinetic data for each subject are available as ASCII files at Figshare (see the 'Results' for details). An exemplary Matlab code is also available in the Supplemental Information.

## Statistical analysis of the processed data

The normality and homogeneity of variances assumptions of the dependent variables were tested using the Bartlett's test. Either one-way ANOVAs or Kruskal-Wallis tests were conducted to examine the effect of running speed on gait-biomechanics variables when the dependent variables did or did not address the assumptions, respectively, at a significance level of 0.05. Whenever a main effect was observed, *post-hoc* analysis was conducted using $t$-tests with Bonferroni adjustments to minimize type I statistical errors. A multinomial logistic regression analysis was performed to determine the effect of running speed (the predictor) on foot-strike patterns (the categorical response variable). The statistical calculations were performed in Matlab 9.0 2016a (Mathworks Inc., Natick, MA, USA) using the Statistics toolbox.

# RESULTS

Both the raw and processed data and a metadata file are available at Figshare (DOI: 10.6084/m9.figshare.4543435); the data is stored in ASCII (text) format with tab-separated columns that can be downloaded as a single compressed file that is made available under the CC-BY license (https://creativecommons.org/licenses/by/4.0/).

## Raw data

The raw data are stored in both c3d and text file formats. The c3d file format is a flexible binary file containing all the unprocessed data from a captured trial. This file format is supported by the main motion-capture manufacturers and other biomechanics software programs (e.g., Visual 3D). Although the Cortex software program, which was used to collect data, does offer the capability of processing and analyzing data, the raw files available in the present dataset contain only 3D, raw marker coordinates and transformed force signals (i.e., forces (N) transformed into the laboratory coordinate system).

**Table 2  Description of the 12 file names per subject in the Running Biomechanics Data set.**

| Type | File name | Description |
|---|---|---|
| C3D | RBDSxxxstatic.c3d | Standing calibration trial data |
| C3D | RBDSxxxrunT25.c3d | Markers and forces data for running at 2.5 m/s |
| C3D | RBDSxxxrunT35.c3d | Markers and forces data for running at 3.5 m/s |
| C3D | RBDSxxxrunT45.c3d | Markers and forces data for running at 4.5 m/s |
| ASCII | RBDSxxxstatic.txt | Standing calibration trial data |
| ASCII | RBDSxxxrunT25markers.txt | Markers data for running at 2.5 m/s |
| ASCII | RBDSxxxrunT35markers.txt | Markers data for running at 3.5 m/s |
| ASCII | RBDSxxxrunT45markers.txt | Markers data for running at 4.5 m/s |
| ASCII | RBDSxxxrunT25forces.txt | Forces data for running at 2.5 m/s |
| ASCII | RBDSxxxrunT35forces.txt | Forces data for running at 3.5 m/s |
| ASCII | RBDSxxxrunT45forces.txt | Forces data for running at 4.5 m/s |
| ASCII | RBDSxxxprocessed.txt | Time-normalized kinematics and kinetics data for all speed conditions |

The c3d files contain both the raw marker trajectories and force signals (both raw analog signals (V) and transformed signals (N)) in a single file. In contrast, separate text files were created for markers and forces signals, as these signals were sampled at 150 Hz for kinematics and 300 Hz for kinetics data. In addition, there is one c3d and one ASCII text file containing only marker trajectories corresponding to one second of the standing, anatomical calibration data of each subject. Finally, the average time-normalized kinematics (joint angles) and kinetics (joint torques, joint powers, and GRF) data for each subject are available in the ASCII file. Hence, as the running trials were performed on an instrumented treadmill at three distinct gait speeds (2.5 m/s, 3.5 m/s, and 4.5 m/s), there are four c3d files and eight text files per subject. Table 2 describes the file-naming convention used for the raw dataset.

The abbreviation RBDS in the file names stands for "Running Biomechanics Dataset" and xxx refers to the subject's identification number (e.g., the first subject is 001). The c3d files can be analyzed using the Visual 3D software program and the supplied model template file (.mdh) along with the pipeline command files (.v3s) or other software, including Mokka (http://biomechanical-toolkit.github.io/mokka/). The ASCII files with marker trajectories contain 97 columns, with the first column containing the recording time (in seconds) and the remaining 96 columns being the $X$, $Y$, and $Z$ coordinates (in millimeters) of the markers in the laboratory-coordinate system. The number of columns varied between the static and running trials (145 vs. 97, respectively) since the markers deemed solely anatomical were removed before the running trials (see the 'Methods'). The headers of the marker files contain the markers' labels (except for the first column, which is elapsed time) and are consistent with the "label" column in Table 1. In turn, the columns of the forces ASCII files are comprised of the forces, center of pressure, and free moment about the vertical axis measured by the instrumented treadmill and transformed on to the laboratory-coordinate system. Each force file has the following header: Time [s], Fx [N], Fy [N], Fz [N], COPx [mm], COPy [mm], COPz [mm], and Ty [Nm], followed by data in 9,000 rows and eight columns with six-digit numerical precision.

## Metadata

One file named RBDSinfo (in both .txt and .xlsx formats) is supported with metadata, demographics, running characteristics, foot-strike patterns, muscle-strength and flexibility measurements for each subject. Below is the coding for the metadata. The first word identifies the name of the column in the header.

1. **Subject**: number of subjects (from one to 28).
2. **Filename**: names of files, including format extensions. Table 2 provides descriptions of the files.
3. **Age**: subject's age in years.
4. **Height**: subject's height in centimeters, measured with a calibrated stadiometer.
5. **Mass**: subject's body mass in kilograms, measured with a calibrated scale.
6. **Gender**: subject's gender (M or F).
7. **Dominance**: answer to the question "What leg would you use if you had to kick a ball (R or L)?".
8. **Level**: answer to multiple-choice question about self-assessed level of running performance (only recreational; recreational competitive; competitive; elite).
9. **Experience**: number of months of regular running practice (at least three times/week).
10. **SessionsPerWk**: number of running training sessions per week.
11. **Treadmill**: number of treadmill running training sessions per week.
12. **Asphalt**: number of running training sessions on an asphalt surface per week.
13. **Grass**: number of running training sessions on a grass surface per week.
14. **Trail**: number of trail running training sessions per week.
15. **Sand**: number of running training sessions on sand per week.
16. **Concrete**: number of running training sessions on a concrete surface per week.
17. **SurfaceAlt**: number of running training sessions per week on other surfaces not listed previously.
18. **RunGrp**: whether the subjects participated in a running training group, as self-declared (Yes or No).
19. **Volume**: weekly running training volume (kilometers/week).
20. **Pace**: average running pace in the shortest long-distance running race (minutes/kilometer).
21. **RaceDist**: running race distance participated in recently, as self-declared (in kilometers).
22. **Injury**: answer to the question "Have you experienced any injury or pain that has significantly affected your running practice?" (Yes or No).
23. **InjuryLoc**: anatomical location of the most recent injury.
24. **DiagnosticMed**: answer to the question "Was this injury medically diagnosed?" (Yes or No).
25. **Diagnostic**: diagnosis of running-related injury, as self-declared.
26. **InjuryOnDate**: approximate date of onset of injury symptoms, as self-declared (dd/mm/yyyy).
27. **ShoeSize**: size of running shoes, as self-declared (US standard).
28. **ShoeBrand**: preferred running shoe manufacturer, as self-declared.
29. **ShoeModel**: model of running shoes, as self-declared.

30. **ShoePairs**: current number of pairs of running shoes, as self-declared.

31. **ShoeChange**: answer to the following multiple choice question "How often do you replace your running shoes?" (less than 6 months; between 7 months and 1 year; between 1 and 2 years; more than 2 years).

32. **ShoeComfort**: comfort rating of their current running shoes, as self-declared on a 10-point rating scale. An average rating score was calculated if they had more than one pair of shoes.

33. **ShoeInsert**: type of foot insert (if any) worn in their running shoes (off-the-shelf insoles; orthotics; taping; none).

34. **RFSI25**: right foot-strike pattern (rearfoot, midfoot, or forefoot) while running at 2.5 m/s (see description in 'Methods').

35. **RFSI35**: right foot-strike pattern (rearfoot, midfoot, or forefoot) while running at 3.5 m/s (see description in 'Methods').

36. **RFSI45**: right foot-strike pattern (rearfoot, midfoot, or forefoot) while running at 4.5 m/s (see description in 'Methods').

37. **LFSI25**: left foot-strike pattern (rearfoot, midfoot, or forefoot) while running at 2.5 m/s (see description in 'Methods').

38. **LFSI35**: left foot-strike pattern (rearfoot, midfoot, or forefoot) while running at 3.5 m/s (see description in 'Methods').

39. **LFSI45**: left foot-strike pattern (rearfoot, midfoot, or forefoot) while running at 4.5 m/s (see description in 'Methods').

40. **RThomas:** angle of the right thigh relative to the horizontal during the Thomas' test measured with a magnetic angle locator, in degrees (see description in 'Methods'). Positive and negative values represent the thigh below and above a line parallel to the therapeutic bench.

41. **LThomas:** angle of the left thigh relative to the horizontal during the Thomas' test measured with a magnetic angle locator, in degrees (see description in 'Methods'). Positive and negative values represent the thigh below and above a line parallel to the therapeutic bench.

42. **ROber**: angle of the right thigh relative to the horizontal during the Ober's test measured with a magnetic angle locator, in degrees (see description in 'Methods'). Positive and negative values represent the thigh below and above a line parallel to the therapeutic bench.

43. **LOber**: angle of the left thigh relative to the horizontal during the Ober's test measured with a magnetic angle locator, in degrees (see description in 'Methods'). Positive and negative values represent the thigh below and above a line parallel to the therapeutic bench.

44. **RHIPABD**: average maximal isometric voluntary contraction (MIVC) of the right hip abductors measured by a hand-held dynamometer (HHD) in kilograms (see *Fukuchi et al., 2014*).

45. **LHIPABD**: MIVC of the left hip abductors measured by an HHD in kilograms.

46. **RHIPEXT**: MIVC of the right hip extensors measured by an HHD in kilograms.

47. **LHIPEXT**: MIVC of the left hip extensors measured by an HHD in kilograms.

48. **RHIPER**: MIVC of the right hip external rotators measured by an HHD in kilograms.
49. **LHIPER**: MIVC of the left hip external rotators measured by an HHD in kilograms.
50. **RHIPIR**: MIVC of the right hip internal rotators measured by an HHD in kilograms.
51. **LHIPIR**: MIVC of the left hip internal rotators measured by an HHD in kilograms.

## Processed data

The processed data comprise average 3D time-normalized joint angles (hip, knee, and ankle), joint torques, and GRFs along with the joint powers at the sagittal plane. These data have six-digit precision (except the percentage column, which is an integer number), and they are stored in a single tab-separated ASCII text file. The processed data were stored such that their columns consisted of the following variables: Gait cycle ($101 \times 1$) (percentage), 3D joint angles ($101 \times 9$) (degrees), 3D joint moments ($101 \times 9$) (Nm/kg), 3D GRFS ($101 \times 3$) (N/kg), and scalar joint powers ($101 \times 3$) (W/kg). The numbers within parentheses represent the dimensions of the matrix of data (number of rows and columns), considering only one gait speed and one lower limb. Table 3 displays the arrangement of the first 25 columns of processed data stored in the ASCII files. Since there are three gait speeds (2.5 m/s, 3.5 m/s, and 4.5 m/s) and two lower limbs (right and left), the resultant matrix has 101 rows and 145 columns (144 columns of biomechanics data plus one column of gait-cycle percentage data).

## Effect of gait speed on running biomechanics

To study the effects of gait speed on running biomechanics, we compared the kinematics and kinetics running patterns of the subjects across gait speeds. Note that the subjects had an accommodation period at each running speed (see Protocol in the 'Methods') before the data were recorded. Figures 3 and 4 show the average pattern (across subjects) of the lower-extremity 3D joint angles and the joint torques curves, respectively. Figure 5 shows the average GRF curves in the medial-lateral, anterior-posterior, and vertical directions along with the lower extremity joint power curves at the sagittal plane. Overall, an increase could be observed in the magnitude of both kinematics (joint angles) and kinetic (torques, GRFs, and powers) variables following an increase in gait speed.

For a more specific, quantitative examination of the effects of gait speed, Table 4 shows the results of the descriptive and inferential statistical analyses. Six of the 24 variables did not meet the assumptions for ANOVA and they were compared using Kruskal-Wallis tests. Figure 6 shows repeated-measures plots with the distribution of the subjects' data across running speeds for all variables that had significant effects. The results of the *post-hoc* analyses are indicated whenever a significant difference was found in the pairwise comparisons. The running-speed conditions of 2.5 m/s, 3.5 m/s, and 4.5 m/s are defined hereafter as V1, V2, and V3, respectively.

### Gait kinematics

A main effect of speed was observed for both stride length and cadence but not for stride width (Table 4). The *post-hoc* analyses revealed that both stride length and cadence increased significantly for all conditions tested (Fig. 6 and Table 4).

Fukuchi et al. (2017), *PeerJ*, DOI 10.7717/peerj.3298

**Table 3 Arrangement of processed data in the first 25 columns, comprising the joint angles, joint torques, GRFs, and joint powers for one speed condition and one lower limb.**

| Cycle (%) | Joint angle (°) | | | | | | | | | Joint torque (Nm/kg) | | | | | | | | | GRF (N/kg) | | | Joint power (W/kg) | | |
|---|---|---|---|---|---|---|---|---|---|---|---|---|---|---|---|---|---|---|---|---|---|---|---|---|
| | HIP | | | KNEE | | | ANKLE | | | HIP | | | KNEE | | | ANKLE | | | | | | HIP | KNEE | ANKLE |
| | X | Y | Z | X | Y | Z | X | Y | Z | X | Y | Z | X | Y | Z | X | Y | Z | X | Y | Z | | | |
| 1 | 2 | 3 | 4 | 5 | 6 | 7 | 8 | 9 | 10 | 11 | 12 | 13 | 14 | 15 | 16 | 17 | 18 | 19 | 20 | 21 | 22 | 23 | 24 | 25 |

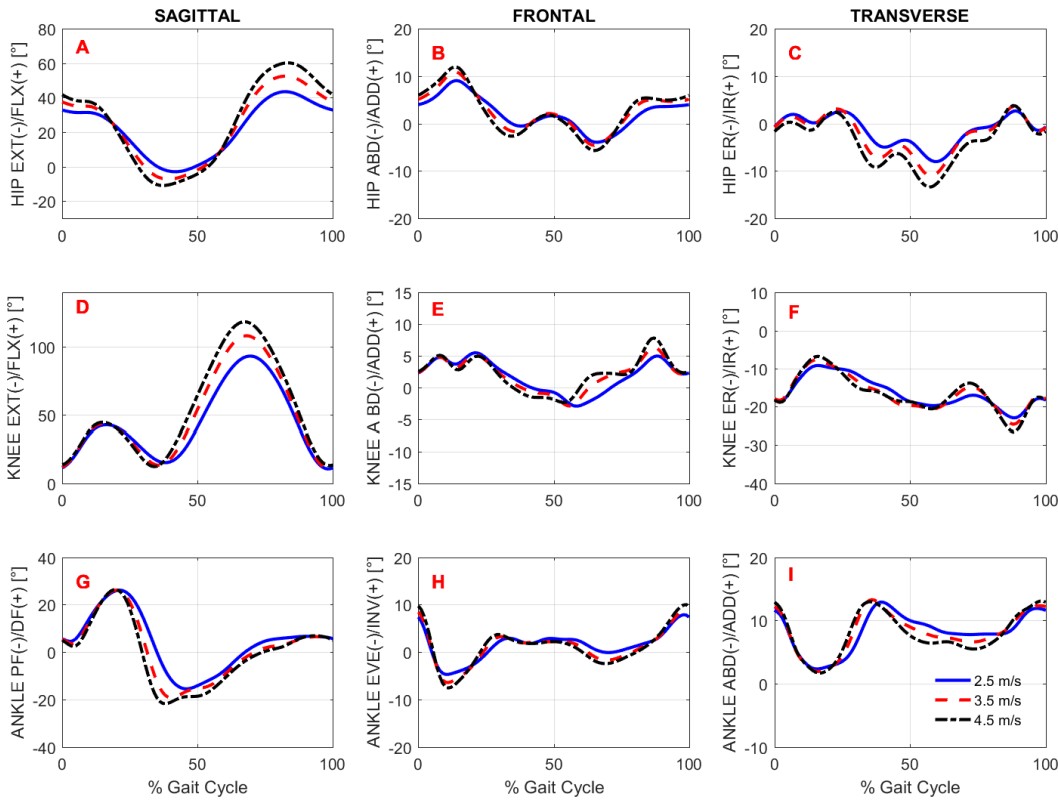

**Figure 3  Lower extremity joint angles.** Ensemble time series of 3D hip (A–C), knee (D–F) and ankle (G–I) joint angles across participants during treadmill running at 2.5 m/s, 3.5 m/s, and 4.5 m/s.

Overall, the lower extremity joint angles were affected by running speed, since main effects were observed in the peak angles of the hip, knee, and ankle joints, except for ankle dorsiflexion (Fig. 3 and Table 4). The maximal angles of hip extension, hip flexion, and knee flexion differed across all possible comparisons. Compared to V1, the relative increases of these variables at V2 and V3 were, respectively, in degrees: hip extension (4.4, 8.2), hip flexion (9.0, 16.8), and knee flexion (15.1, 25.6). In contrast, when V2 and V3 were compared with each other, the peak ankle plantar flexion was not altered. The maximum ankle eversion angle also exhibited higher values at higher speeds; however, the *post-hoc* analysis revealed that this variable differed only when V1 and V3 were compared.

The foot-strike pattern distribution in V1, V2, and V3 were, respectively, rearfoot strikers (RFS): 68%, 68%, 61%; midfoot strikers (MFS): 14%, 18%, 21%; and forefoot strikers (FFS): 18%, 14%, 18%. Contrary to our hypothesis, the coefficients of the multinomial logistic regression model revealed that foot-strike patterns were not affected by gait speed. The probability of changing an RFS pattern (reference category) to either an FFS pattern ($\beta_0 = -1.56 \pm 1.32$; $IC_0$ [−4.14; 1.02]; $p_0 = 0.236$; $\beta_1 = 0.05 \pm 0.37$; $IC_1$ [−0.66; 0.77]; $p_1 = 0.882$) or an MFS pattern ($\beta_0 = -2.21 \pm 1.33$; $IC_0$ [−0.45; 0.96]; $p_0 = 0.477$; $\beta_1 = 0.27 \pm 0.36$; $IC_1$ [−0.45; 0.96]; $p_1 = 0.477$) remained unaltered by any increment in gait speed. The term $\beta_0 \pm$ se includes the coefficient and standard error (se) of the constant; $\beta_1 \pm$

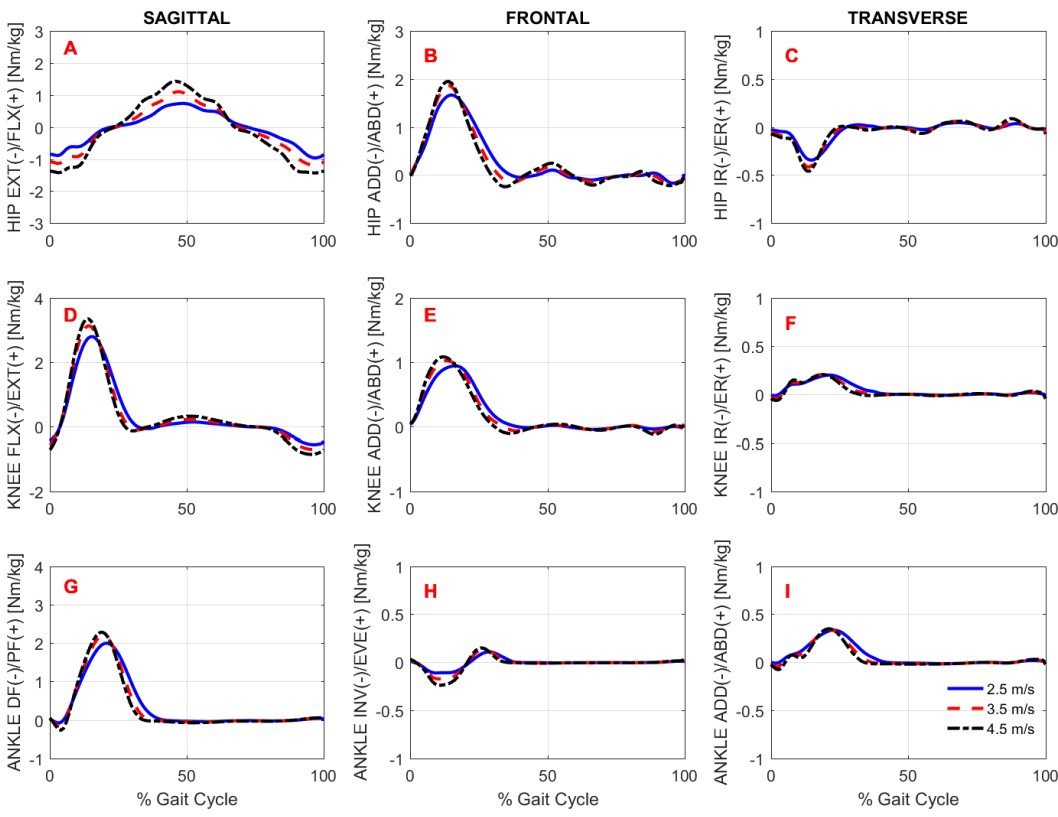

**Figure 4 Lower extremity joint torques.** Ensemble time series of 3D hip (A–C), knee (D–F) and ankle (G–I) joint torques across participants during treadmill running at 2.5 m/s, 3.5 m/s, and 4.5 m/s.

se includes the coefficient and se of the predictor (gait speed); IC is the confidence interval of the coefficients; and *p* is the associated *p*-value.

### Gait kinetics

Overall, there was an effect of running speed on joint torques (in Nm/kg), joint work (in J/kg), and GRF variables (in Ns/kg), as can be seen in Figs. 4–6 and Table 4. Compared to V1, the percentage of increase in hip extensor and flexor torque peaks at higher running speeds (V2 and V3), compared to V1, were 0.31 and 0.61; and 0.37 and 0.70, respectively. In addition, compared to V1, at V2 and V3, the knee extensor torque increased 0.34 and 0.57, respectively, and the ankle extensor torque increased 0.21 and 0.31, respectively. In contrast, no difference was found when V2 and V3 were compared to each other. In addition, a main effect of running speed was found at the ankle flexor torque but only when V1 and V3 were compared. Contrary to our hypothesis, the knee abduction impulse (area under the torque-time curve) was not affected by running speed (Table 4). Compared to V1, the GRF propulsive, GRF braking, and GRF vertical impulses increased 0.07, 0.15 and 0.08, respectively, at V2 and 0.13, 0.22, and 0.15, respectively, at V3 (Table 4). The *post-hoc* analyses found effects for all conditions tested in the aforementioned variables, except for the GRF braking impulses between V2 and V3. The hip and ankle positive works were affected in all tested conditions. Compared to V1, they were increased 0.69 and 0.14,

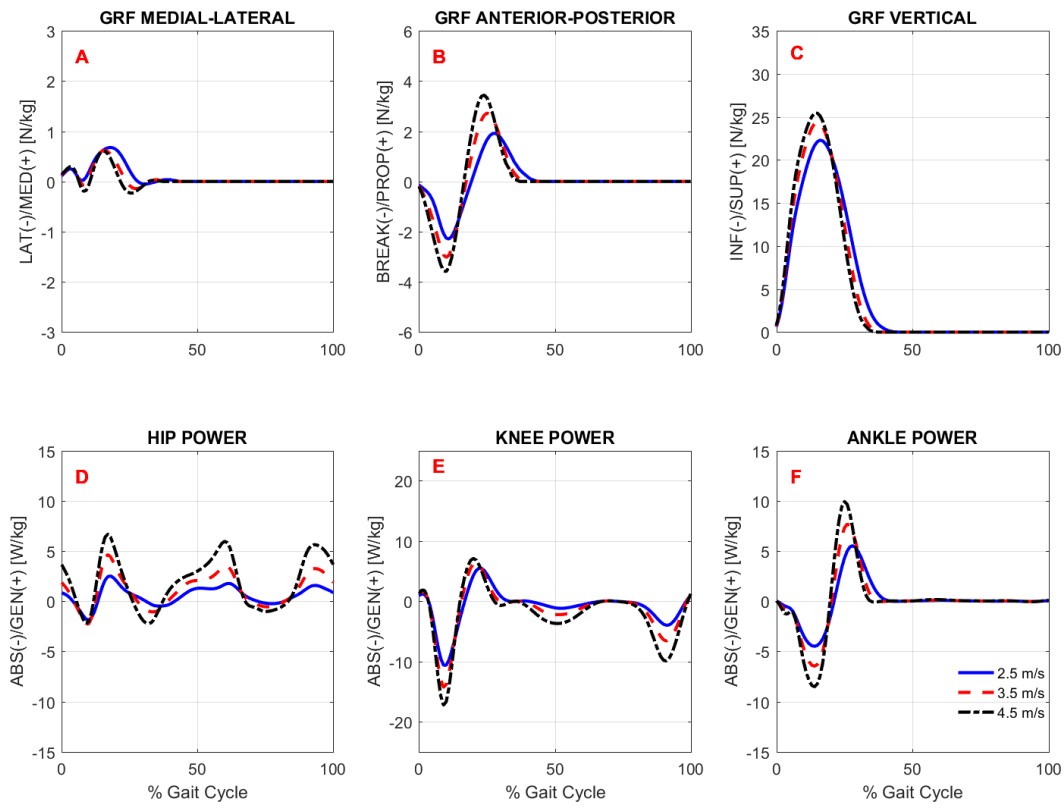

**Figure 5** **Ground reaction forces and joint powers.** Ensemble time series of 3D GRF forces (A–C) and hip (D), knee (E), and ankle (F) powers at the sagittal plane across participants during treadmill running at 2.5 m/s, 3.5 m/s, and 4.5 m/s.

respectively, at V2 and 1.62 and 0.32, respectively, at V3. In contrast, the knee positive work remained constant when V2 and V3 were compared to each other. However, compared to V1, it increased 0.17 and 0.23, respectively, at V2 and V3. Compared to V1, the hip, knee, and ankle negative joint work increased 0.16, 0.65, and 0.19, respectively, at V2 and 0.39, 1.53, and 0.39, respectively, at V3 (Table 4). The distributions of positive and negative work across lower extremity joint are shown in Fig. S1.

## DISCUSSION

This study presented a publicly available dataset on regular runners' gait biomechanics (kinematics and kinetics), demographics, running habits, muscle strength, flexibility, and foot-strike patterns. Biomechanical datasets have begun to be accessible in public repositories recently (see *Moore, Hnat & Van den Bogert, 2015*; *Santos & Duarte, 2016*, and the references therein). However, these datasets primarily consist of walking and posture data. We provided both raw data (in two formats: .c3d and .txt) and processed data for reuse, along with metadata and other files that can be used to reproduce the processed data. In addition, the study examined the effect of running speed on selected gait variables commonly associated with running injuries and running economy. The study observed that running speed significantly affected lower-extremity kinematics and kinetics.

**Table 4 Descriptive and inferential statistics for the kinematic and kinetic variables of 28 subjects during treadmill running at 2.5 m/s, 3.5 m/s, and 4.5 m/s.** In the results of the *post-hoc* multiple comparisons with Bonferroni adjustments, 0 indicates no difference, and 1 indicates a significant difference in the pairwise comparison. The symbol # indicates variables compared using the Kruskal-Wallis test.

| Variables | 2.5 m/s | 3.5 m/s | 4.5 m/s | Mean relative difference | | ANOVA | | Multiple comparisons | | |
|---|---|---|---|---|---|---|---|---|---|---|
| | Mean ± SD | Mean ± SD | Mean ± SD | V2–V1 | V3–V1 | $F$ or $\chi^2$ | $p$-value | 'V1V2' | 'V1V3' | 'V2V3' |
| Stride length (m) | 1.86 ± 0.11 | 2.46 ± 0.15 | 2.96 ± 0.20 | 0.60 | 1.10 | 335.39# | 0.000 | 1 | 1 | 1 |
| Cadence (strides per minute) | 80.82 ± 4.63 | 85.68 ± 5.27 | 91.74 ± 6.69 | 4.86 | 10.92 | 26.72 | 0.000 | 1 | 1 | 1 |
| Stride width (m) | 0.10 ± 0.02 | 0.09 ± 0.02 | 0.08 ± 0.02 | −0.01 | −0.01 | 2.60 | 0.080 | – | – | – |
| Max hip flx angle (°) | 43.75 ± 6.06 | 52.76 ± 5.75 | 60.50 ± 6.06 | 9.01 | 16.76 | 55.48 | 0.000 | 1 | 1 | 1 |
| Max hip ext angle (°) | −3.58 ± 4.85 | −7.95 ± 4.58 | −11.75 ± 4.78 | 4.37 | 8.18 | 20.90 | 0.000 | 1 | 1 | 1 |
| Max knee flx angle (°) | 93.52 ± 10.36 | 108.68 ± 10.65 | 119.12 ± 10.37 | 15.15 | 25.59 | 42.37 | 0.000 | 1 | 1 | 1 |
| Knee ABD impulse (Nms) | 0.20 ± 0.06 | 0.20 ± 0.06 | 0.20 ± 0.06 | 0.00 | 0.01 | 0.10 | 0.905 | – | – | – |
| Max ankle DF angle (°) | 26.36 ± 2.93 | 26.54 ± 2.49 | 26.79 ± 2.51 | 0.18 | 0.43 | 0.19 | 0.831 | – | – | – |
| Max ankle PF angle (°) | −16.62 ± 5.50 | −20.47 ± 4.71 | −23.17 ± 4.72 | 3.84 | 6.54 | 12.15 | 0.000 | 1 | 1 | 0 |
| Max eversion angle (°) | −4.91 ± 2.74 | −6.59 ± 2.99 | −7.81 ± 3.59 | 1.68 | 2.90 | 6.07 | 0.004 | 0 | 1 | 0 |
| Max hip flx torque (Nm/kg) | 0.78 ± 0.11 | 1.15 ± 0.14 | 1.49 ± 0.19 | 0.37 | 0.70 | 150.42# | 0.000 | 1 | 1 | 1 |
| Max hip ext torque (Nm/kg) | −1.06 ± 0.14 | −1.37 ± 0.19 | −1.67 ± 0.21 | 0.31 | 0.61 | 76.50 | 0.000 | 1 | 1 | 1 |
| Max knee ext torque (Nm/kg) | 2.84 ± 0.45 | 3.18 ± 0.50 | 3.41 ± 0.47 | 0.34 | 0.57 | 10.28 | 0.000 | 1 | 1 | 0 |
| Max ankle PF torque (Nm/kg) | 2.03 ± 0.22 | 2.23 ± 0.23 | 2.34 ± 0.25 | 0.21 | 0.31 | 13.11 | 0.000 | 1 | 1 | 0 |
| Max ankle DF torque (Nm/kg) | −0.14 ± 0.11 | −0.23 ± 0.16 | −0.32 ± 0.20 | 0.09 | 0.18 | 8.61# | 0.000 | 0 | 1 | 0 |
| Hip pos work (J/kg) | 0.80 ± 0.20 | 1.49 ± 0.30 | 2.43 ± 0.39 | 0.69 | 1.62 | 195.16# | 0.000 | 1 | 1 | 1 |
| Hip neg work (J/kg) | −0.27 ± 0.09 | −0.42 ± 0.12 | −0.66 ± 0.22 | 0.16 | 0.39 | 44.24# | 0.000 | 1 | 1 | 1 |
| Knee pos work (J/kg) | 0.69 ± 0.17 | 0.86 ± 0.19 | 0.92 ± 0.23 | 0.17 | 0.23 | 10.08 | 0.000 | 1 | 1 | 0 |
| Knee neg work (J/kg) | −1.50 ± 0.18 | −2.15 ± 0.24 | −3.03 ± 0.30 | 0.65 | 1.53 | 274.07# | 0.000 | 1 | 1 | 1 |
| Ankle pos work (J/kg) | 0.64 ± 0.10 | 0.78 ± 0.10 | 0.95 ± 0.15 | 0.14 | 0.32 | 50.90 | 0.000 | 1 | 1 | 1 |
| Ankle neg work (J/kg) | −0.58 ± 0.13 | −0.77 ± 0.14 | −0.96 ± 0.15 | 0.19 | 0.39 | 52.86 | 0.000 | 1 | 1 | 1 |
| GRF brak impulse A–P (Ns/kg) | −0.34 ± 0.10 | −0.49 ± 0.13 | −0.56 ± 0.14 | 0.15 | 0.22 | 22.16 | 0.000 | 1 | 1 | 0 |
| GRF prop impulse A–P (Ns/kg) | 0.27 ± 0.07 | 0.33 ± 0.09 | 0.40 ± 0.12 | 0.07 | 0.13 | 12.75 | 0.000 | 1 | 1 | 1 |
| GRF pos impulse vertical (Ns/kg) | 5.04 ± 0.09 | 5.12 ± 0.10 | 5.19 ± 0.10 | 0.08 | 0.15 | 17.44 | 0.000 | 1 | 1 | 1 |

Even though there has been an increased number of publications about running biomechanics, there is a scarcity of publicly available running data sets which hampers the dissemination of biomechanics data and prevents a wider use of published data. To help address this problem, we presented a data set of running biomechanics of regular runners. Compared to other available gait data sets, the present set includes both raw and processed data in various files formats. In addition, running biomechanics at different controlled gait speeds, from multiple gait cycles, considering both limbs and both kinematics and kinetics data are provided. Furthermore, a metadata file is included with the necessary information pertaining to each file and participant of the study to enhance the dissemination and wide use of the data. There are certainly other data sets previously published that fulfil the same recommendations desired to disseminate and enhance the reuse of data, however, to our knowledge, none of them assessed running biomechanics (*Moore, Hnat & Van den Bogert, 2015*).

When running speed was increased, the participants adopted longer stride length and greater stride frequency. The stride length increased to a greater extent than stride

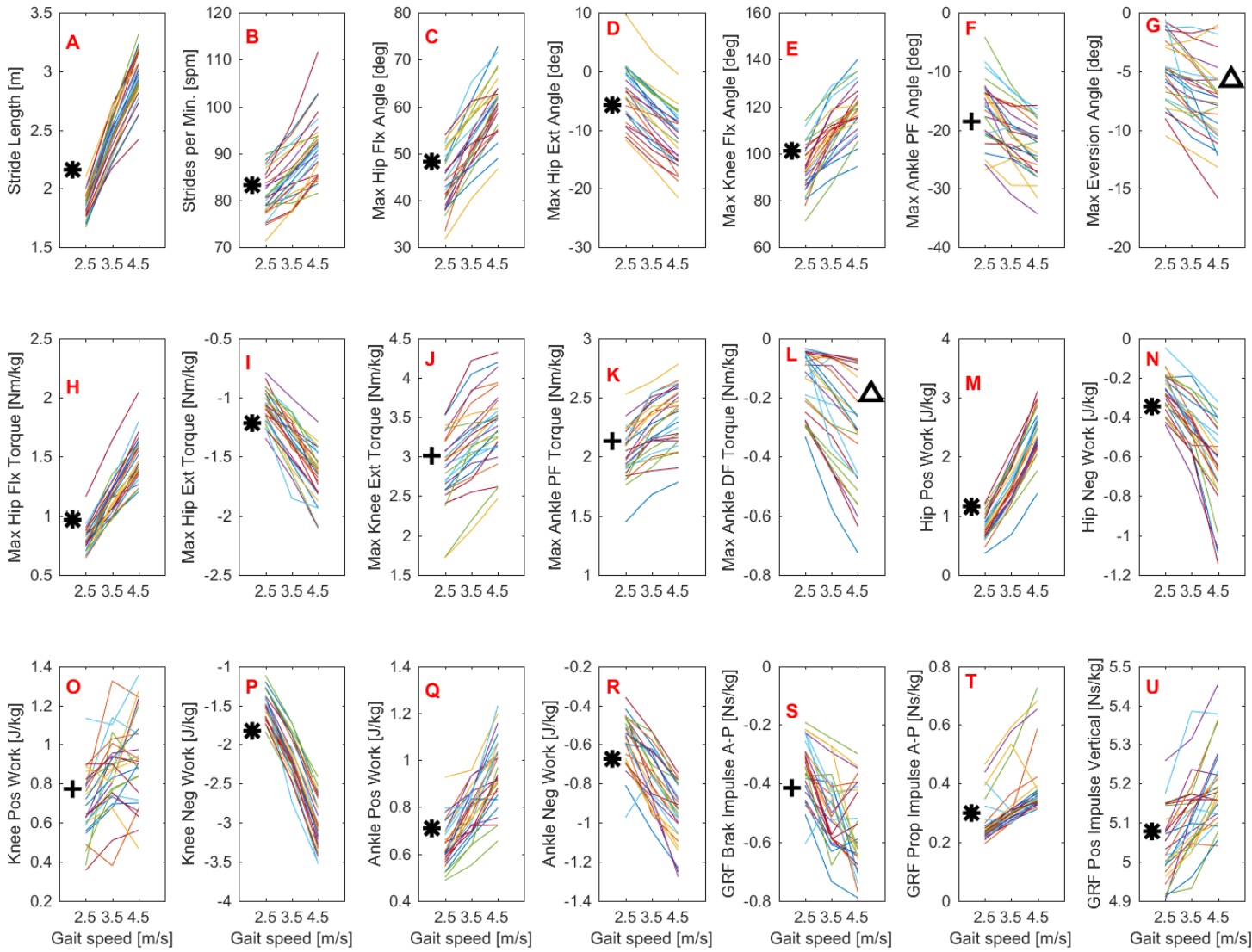

**Figure 6 Kinematic and kinetic values distribution across the range of running speeds.** Plots highlighting the distribution of the 28 subjects' values across running speeds in the kinematic (A–G) and kinetic (H–U) variables. Significant differences in the post-hoc analyses are indicated by the symbols ∗, +, and Δ. ∗Significant difference in all pairwise comparisons. +Significant difference between 2.5 m/s vs. 3.5 m/s and between 2.5 m/s vs. 4.5 m/s. ΔSignificant difference only between 2.5 m/s vs. 4.5 m/s.

frequency (39.3% *vs.* 9.3% on average across the gait speeds). *Dorn, Schache & Pandy (2012)* reported similar results, in which the stride length increased at higher rates at long-distance running speeds on the ground. *Schache et al. (2014)* stated that the results of *Dorn, Schache & Pandy (2012)* suggest that to increase their running speed, humans choose to push the ground more forcefully rather than more frequently, particularly, at slow-to-medium pace running. This is also compatible with the higher values we found for the GRF horizontal propulsive impulse. The peak flexion and extension values of the hip, knee, and ankle angles also increased at higher running speeds, except for the peak ankle dorsiflexion angle. These results were expected, since the runners had to use larger strides, and thus greater joint displacement, to cope with higher running speeds. Similar

results have been reported elsewhere (*Dorn, Schache & Pandy, 2012*). The present study observed no change in the foot-strike patterns as running speeds increased. Although there is a general understanding that the point of contact shifts from the rear toward the anterior part of the foot as running speed increases (*Cheung et al., 2017*), this may not be true for speeds below 5 m/s (*Breine et al., 2014*; *Hatala et al., 2013*), i.e., within the range adopted in the present study. This contrasting evidence across studies highlights the fact that the relationship between running speed and foot-strike patterns is complex and needs to be examined further, particularly considering long distance running pace. Several factors may explain the differences in the findings across studies such as the range of running speeds, running surface (treadmill vs. overground), shoes, and different equipment or measurement methods used to quantify foot-strike patterns. Therefore, these factors need to be considered in future studies. Foot eversion (pronation) has long been associated with running injuries; however, there is limited understanding of whether this is influenced by running speed. Although the present study observed a significant main effect of running speed on the peak ankle eversion angle, the *post-hoc* analysis revealed differences only when speeds of 2.5 m/s and 3.5 m/s were compared. Similar results have been reported in recreational runners during treadmill running at comfortable speeds (*Munoz-Jimenez et al., 2015*).

In general, the lower extremity joint torques and joint work were also affected by increased running speed. In particular, the hip torques (both flexion and extension), hip work (both positive and negative), and ankle positive work were all significantly affected by running speed in all conditions tested. The important contribution of the ankle plantar flexors to generating propulsive force and thereby increasing gait speed has been investigated both experimentally and through simulation studies (*Hamner & Delp, 2013*; *Schache et al., 2014*). Regarding hip-joint loading, there is evidence that the participation of the hip in power generation increases non-linearly as a function of running speed (*Schache et al., 2011*). Similar behavior was observed in the present study when the rate of increase in hip power generation was not constant compared to the work and torque at the knee and ankle (see relative increase in Table 4). This finding may be explained by the fact that the work done by the hip muscles to accelerate the leg during the swing phase increases at a faster rate to move the leg forward more rapidly. The knee extension torque and positive work were also affected by running speed, but to a lesser extent than the hip and ankle, since they remained unaltered when speeds of 3.5 m/s *vs.* 4.5 m/s were compared. In line with our hypothesis, the GRF horizontal and vertical impulses were affected by running speed. These results were expected, since the leg must apply higher impulses to the surface to increase gait speed. In particular, the increment of the GRF vertical impulse with increased speed was only about 2.5% on average, although it was statistically significant.

The present study presented new findings and partly addressed some limitations observed in previous studies, including failing to consider both the stance and swing phases of the gait cycle, small sample sizes, limited joints and set of variables (e.g., only kinematics or kinetics); however, other limitations persist. The use of discrete variables from time-series curves may be too simplistic to deal with the complex nature of gait-biomechanics data (*Lai, Begg & Palaniswami, 2009*). Even the area under the force-time and power-time

curves may not be sufficient to capture the overall pattern of the subjects. While our results seem to be in agreement with those of a handful of other studies, the potential presence of soft-tissue artifact must be acknowledged, even though all experimental procedures were performed carefully to minimize errors from this source. The data were collected while the subjects ran on an instrumented treadmill which certainly was not the first choice of practice environment for most of runners in this study (see metadata file). Therefore, the adopted testing procedures may not be representative of the training and race conditions experienced by the runners and caution should be taken when generalizing from the present findings. In particular, the foot strike index obtained on the treadmill may not necessarily be the same as in overground condition. Nevertheless, the treadmill offers the possibility of controlling gait speed while collecting sufficient trials (footfalls) to represent each subject's pattern. Finally, the subjects wore standard neutral shoes rather than their own shoes. Whether this is an issue is unknown, however we acknowledge that by introducing "new" shoes may require longer familiarization time than what was allowed for the subjects.

Despite the fact that the present data set have many applications in future studies, the extent of its use is limited by some factors. Although standardized and detailed described within the manuscript, the data collection procedures may differ from other laboratories with respect, including but not limited to the marker set protocol, the running shoes, the selected gait speed, the treadmill condition. Hence, caution should be taken when combining this data, particularly when comparing the present data set with others. In addition a Visual 3D biomechanics model (mdh file) is supplied and it can be reused or reproduced in other data sets as long as the same marker set protocol is used. With regards to the treadmill condition, as discussed earlier, caution should be taken when comparing the results with sets of data using different conditions (i.e., overground) or even with different treadmill models. Finally, there is an emerging field of research on wearable sensors to monitor daily life activities including gait that must be acknowledged (*Picerno, 2017*). Whilst the validity and reliability of this technology are not comparable to the data, particularly for non-sagittal movement, obtained in biomechanics laboratories using motion capture systems and force plates, the use of these sensors enhance the ecological validity of the findings since they allow the individuals to run freely in their natural environment and training conditions.

The raw dataset provided by this study allows the reuse of this set to test novel approaches to address some of the present limitations. Although a great deal of effort was made to collect and prepare the present dataset, it likely contains deviations, as would any dataset. Therefore, caution should be taken when interpreting the results derived from these data.

## CONCLUSIONS

A public dataset of running biomechanics and other data pertaining to running practice has been presented and is available in a public repository. The detailed description of the experimental procedures and the supplied files used for data processing will allow other research groups to generate similar sets of data to expand the current one as well as to reuse them. A number of applications of this dataset can be anticipated, including testing new

methods of reducing data and selecting variables; for educational purposes, and answering specific research questions. With the inclusion of additional subjects, this data set may also serve as reference normative data. In fact, this dataset was useful for addressing the question of whether running speed affects gait biomechanics. The study observed an overall effect of running speed on the kinematic and kinetic variables associated with injuries. In contrast, contrary to our hypothesis, the foot-strike pattern remained unaltered and the eversion angle of the foot was altered only during extreme running speeds. Given the emerging interest in data sharing, there is a need to elaborate standards to present and disseminate gait biomechanics data outlining, among other factors, the minimum set of data required for studying running biomechanics and the potential inclusion of data from wearable sensors.

### Funding

This study was supported by Fundação de Amparo à Pesquisa do Estado de São Paulo from Brazil (#2008/10461-7, #2015/14810-0, #2014/13502-7) and Conselho Nacional de Desenvolvimento Científico e Tecnológico from Brazil (487490/2013-4). The funders had no role in study design, data collection and analysis, decision to publish, or preparation of the manuscript.

### Grant Disclosures

The following grant information was disclosed by the authors:
Fundação de Amparo à Pesquisa do Estado de São Paulo: #2008/10461-7, #2015/14810-0, #2014/13502-7.
Conselho Nacional de Desenvolvimento Científico e Tecnológico: 487490/2013-4.

### Competing Interests

The authors declare there are no competing interests.

### Author Contributions

- Reginaldo K. Fukuchi conceived and designed the experiments, performed the experiments, analyzed the data, contributed reagents/materials/analysis tools, wrote the paper, prepared figures and/or tables, reviewed drafts of the paper.
- Claudiane A. Fukuchi performed the experiments, analyzed the data, contributed reagents/materials/analysis tools, wrote the paper, prepared figures and/or tables, reviewed drafts of the paper.
- Marcos Duarte conceived and designed the experiments, analyzed the data, contributed reagents/materials/analysis tools, wrote the paper, reviewed drafts of the paper.

### Human Ethics

The following information was supplied relating to ethical approvals (i.e., approving body and any reference numbers):

This study was approved by the local ethics committee of the Universidade Federal do ABC (CAAE: 53063315.7.0000.5594) and all experiments were conducted within its facilities.

## Data Availability

A public data set of running biomechanics.

http://demotu.org/datasets/.

## Supplemental Information

Supplemental information for this article can be found online at http://dx.doi.org/10.7717/peerj.3298#supplemental-information.

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
