# Peer review of "A public dataset of running biomechanics and the effects of running speed on lower extremity kinematics and kinetics"

_PeerJ, doi:10.7717/peerj.3298_

## Round 0.1 · original submission · Major Revisions

Reviewers found your paper suitable for PeerJ and I agree with them. However it needs some major amendments. Please see below their comments and suggestion.

·

Basic reporting

The study nicely integrated kinematics and kinetics parameters of human distance running. This well-written paper (IMRAD format) is within the scope of interest of the typical readers and is also properly formatted to PeerJ.

Experimental design

You, authors, reported a study that tested the effects of horizontal speed (2.5, 3.5 and 4.5 m/s) on kinematics and kinetics responses of running. Although interesting and timely, I think the study could be considered more than database, an original study. Currently, expose the database is very common and for some journals, fortunately, is obligatory. The PeerJ is not concerned with potential influence/impact; therefore this ms is not just a comprehensive database.
Using biomechanical measurements, you were able to show how classical (stride length and frequency, ROM, etc.) and modern (non-sagittal kinematics, torque, power, foot-strike pattern) biomechanical parameters were affected by running speed.

Validity of the findings

One strong point of this work is the speeds chosen, very used in endurance training.
As recommended by PeerJ guidelines, I organize my issues by importance:
i) The hypotheses are weak, and you do not respond them in the discussion. Please consider removing them.
ii) The results in percentage should be changed as suggested below.
iii)The Anova test needs include the normality and heteroscedasticity tests. I recommend using a GEE test.
vi) Please consider using a pie chart including the total lower limb mech work, divided by joints (negative and positive work or total) for the 1,2 and 3 conditions.
v) Your introduction needs more detail. I suggest that you improve the description at lines 61- 62 to provide more justification for your study (specifically, you should expand upon the knowledge gap being filled on the basis of your second purpose).

Additional comments

How did you evaluate the dominance?
The table and figures are well presented. But, I’d consider including a table with general sample characterization.
Line 36 – On the one hand.
Line 58 – the development.
Line 202 – Please consider exchanging ‘temporal spatial’ with spatiotemporal.
Line 234 – What do you mean with transformed force signals? Normalized? Moments?
Line 237 – Please consider using two or tow different instead various. 100 and 300Hz, OK?
Line 248 – delete the ,
Line 256 – … respectively) since…
Line 262 – in the ms is COP and in the figshare is M. If you showed the moment, the unit is Nm. Please consider using just one term.
Line 319 - You refer to Cavanagh and Lafortune’ article to determine the foot-strike index, but they assessed the FSI using fixed force platforms. How did you determine the FSI using an instrumented treadmill? Please consider describing these procedures in the ms.
Line 338 – I miss information on ober and thomas test.
Line 380 – … curves in the sagittal…
Line 398 – Here the presentation of percentages is misleading because an increase of 122/229% in hip extension against 21/38% in hip flexion might lead the reader to believe that the hip extension in much more important with increasing speed when is not true. Please consider using the absolute difference (in degrees) or the difference normalized by the ROM joint at 2.5m/s.
Line 419-421 – Again the % differences may lead the reader to think that there were somewhat huge differences (~50-90%) when the differences are relative to basic variation much lesser. Please consider using the absolute difference or the diff related to fixed variation from 2.5m/s.
Again for the joint work.
Line 458 – Due to consistency, consider using only stride frequency.
The discussion on 467-475 is at least ‘forced’/artificial. There is a well-known change on the foot-strike position at high speeds/sprint. From the Breine, and Cheung studies are necessary taking into account in which speeds specifically changed or not the foot-strike pattern.
Line 627 – multiple comparisons

Reviewer 2 ·

Basic reporting

The article fulfils the requirements.

Experimental design

There are two goals listed:
1. to create a comprehensive public data set
evaluating running biomechanics
2. to examine the effect of running speed on selected gait-
biomechanics variables related to both running injuries and running economy.

The first goal is certainly relevant and meaningful.
I congratulate the authors to their extensive data set and their willingness to make it publicly available.

One may want to be more specific about what one means by "comprehensive". There will be certainly some research questions around running biomechanics that one may not be able to answer using the data set. Groups that are analysing data from "wearables" may want to see also data from mobile sensors.

Also the second goal is certainly relevant and meaningful.

The introduction would benefit from some more literature search about the status quo of research about running injuries (e.g. the groups around Benno Nigg, Lieberman, and the many discussions about injuries in the barefoot/minimal footwear community).

Validity of the findings

The first goal has clearly been reached, with the only potential circularity about the precise meaning of "comprehensive". Also the opportunity to develop and test mehtods for data reduction etc. is clearly demonstrated.

The second goal is more problematic. On the one hand the finding is unexpected and to a certain extent, as the authors say, inconsistent with findings from other groups.

The authors list some limitations of the study design but should be more explicit about one major problem with the ecologic vlidity of their findings: runner's typically get their injuries when they are running outdoors in an uncontrolled environment, whereas all the raw data collected comes from highly controlled, artifical indoor/treadmill environment.

The authors are of course not the only ones to have this problems, but there is an interesting field emerging that uses data from mobile sensors in the "real world" of the runners and also social media to collect data about running injuries that may give insights about the risks for running injuries that are independent from details of the biomechanics.

I would recommend to add some sentences that explain this difficulty in more detail.

Other limitation are the following:
- Runners were not all used to run in the shoe provided in the study. Time needed for neuromuscular adaptation may play a role.
- The "fact" that runners change their speed by mainly increase their stride length, less by icreasing step frequency may not necessarily be true for (ultra)long distance runners, showing again the problem of ecologic validity (see e.g. PeerJ PrePrints 2:e497v1 https://doi.org/10.7287/peerj.preprints.497v1).

Additional comments

This is an important article and an important initiative that should stimulate the field to contribute in a similar way. It may be a good idea to start a consensus process to decide what should be a "comprehensive" and or "minimal standard" data set in order to allow research and validation of results. For the latter it may be interesting to think about a concept for a "closed" data set, that is specifically reserverd for validation purposes.

·

Basic reporting

The present manuscript presents a running biomechanics dataset made publicly available based on a population sample of a number of able bodied runners whose biomechanics were collected while running on a a dual belt treadmill.
The focus of the work may be of clear interest for a selected portion of potential PeerJ readers interested in the area of Human movement studies, but a larger portion will potentially benefit from the outcomes of the dataset presentation described in the manuscript.
The relevance of the problem is clearly defined, and its significance in the scientific community is high.
The survey on the state of the art in the introduction is appropriately delimited.
Raw data, both with processed data, are reported as made available through a Figshare website.

Experimental design

The protocol section is accurately described, and details on the data recording process are given.
While the analysis on differences between running speeds is a by-product of this manuscript, results regarding this aspect are reported in full details.

Validity of the findings

no comment. See below.

Additional comments

In overall, I judge the article deserving being published with no substantial modifications to the present form.
I have, however, a number of minor reservations regarding this manuscript, which I would like the authors to fully address before going on with the publishing process. These will be listed in the following:

1) The manuscript makes direct reference, in the abstract and in the introduction (and indirectly in other sections), to the creation of a publicly available dataset. As a matter of fact, the database is already publicly available, with a digital object identifier. In my opinion, this does not heavily impact the nature of the submitted manuscript, but I would suggest the authors to modify the sentences regarding the objective of the study into a more appropriate sentence (e.g. “the purposes of this study were (1) to present the set of raw and processed data on running biomechanics made available publicly at doi…”).

2) The number of individuals who performed the experimental protocol seem to me as rather small ( < 30) for a dataset to be used as reference data in the scientific community. If it will not be possible to add other subjects, this needs to be acknowledged in the discussion section.

3) Regarding the discussion, I have some reservations to the rather unbalanced share between the two purposes of the study: almost the whole discussion focusses in fact on the interpretation of the differences coming from the running speed conditions (which is OK for the purpose #2), but almost no considerations on the nature of the dataset as a whole are reported: I will list here 2-3 questions that the authors may find useful to address to make this point:
a) Are the differences on the captured data from data coming from the literature a possible issue to be considered for the use of the dataset from other authors?
b) Is there a treadmill-specific effect on these data, which would prevent other scientists from comparing their data to the dataset if a different kind of treadmill is used?
c) (this may pertain to the concluding section) Will the dataset be open to additions from other labs, provided the same protocol is used? If yes, which are the necessary steps to be taken into account to perform this? If not, is there a technical limitation for this?

4) I guess this is something that could not be included into the original submission, but it will need to be included before publication: specific reference to the dataset website, doi, and use rights need to be included (if necessary, as an appendix), to let the reader get these data. In this appendix, it would be a very useful plus if the authors made available (not necessarily within the appendix, but by placing a link where this can be downloaded) the coding needed to obtain the summary data presented in the study. This would make it possible for other researchers to compare their own data with the dataset, by considering the same processing implementation.

---

## Round 0.2 · accepted · Accept

Two out of the initial three reviewers accepted the manuscript in its present form and after reading it myself I can accept it without waiting for the third one, who didn't accept my invitation to re-review the manuscript.

·

Basic reporting

In this round of revision, the authors are able to address each one of the issues raised by reviewer.

Experimental design

Experiments are well done and design is acceptable to support authors' conclusion.

Validity of the findings

no comment

Additional comments

I read carefully this updated version of the manuscript and the related rebuttal. The authors had addressed all comments pointed out by me. Congrats!

·

Basic reporting

I am content with the changes made in the revised version of the manuscript.

Experimental design

n.a.

Validity of the findings

n.a.

Additional comments

n.a.